# SEMG Feature Extraction Based on Stockwell Transform Improves Hand Movement Recognition Accuracy

**DOI:** 10.3390/s19204457

**Published:** 2019-10-14

**Authors:** Haotian She, Jinying Zhu, Ye Tian, Yanchao Wang, Hiroshi Yokoi, Qiang Huang

**Affiliations:** 1School of Mechatronical Engineering, Beijing Institute of Technology, Beijing 100081, China; 2Key Laboratory of Biomimetic Robots and Systems, Ministry of Education, Beijing 100081, China; 3Institute of Automation, Chinese Academy of Sciences, Beijing 100190, China; 4Beijing Advanced Innovation Center for Intelligent Robot and System, Beijing 100081, China; 5School of informatics and Engineering, University of Electro-Communications, Tokyo 163-8001, Japan

**Keywords:** surface EMG signal, feature extraction, Stockwell transform, hand movement recognition

## Abstract

Feature extraction, as an important method for extracting useful information from surface electromyography (SEMG), can significantly improve pattern recognition accuracy. Time and frequency analysis methods have been widely used for feature extraction, but these methods analyze SEMG signals only from the time or frequency domain. Recent studies have shown that feature extraction based on time-frequency analysis methods can extract more useful information from SEMG signals. This paper proposes a novel time-frequency analysis method based on the Stockwell transform (S-transform) to improve hand movement recognition accuracy from forearm SEMG signals. First, the time-frequency analysis method, S-transform, is used for extracting a feature vector from forearm SEMG signals. Second, to reduce the amount of calculations and improve the running speed of the classifier, principal component analysis (PCA) is used for dimensionality reduction of the feature vector. Finally, an artificial neural network (ANN)-based multilayer perceptron (MLP) is used for recognizing hand movements. Experimental results show that the proposed feature extraction based on the S-transform analysis method can improve the class separability and hand movement recognition accuracy compared with wavelet transform and power spectral density methods.

## 1. Introduction

Electromyography (EMG) signals are divided into two main groups: needle and surface EMG (SEMG) signals. SEMG signals have been widely used in upper-limb prosthesis control because of their advantages of being convenient and noninvasive [1,2,3,4,5]. However, EMG classification is very complex because of its properties of nonlinearity, nonstationarity and subject dependency [6,7]. There are three main steps for EMG classification: preprocessing, feature extraction, and classifier design. The difficulty in EMG signal classification is extracting a feature vector that is able to classify several motions because the EMG signals are subject-dependent and have noise interference. This step is important because the feature vector has a direct impact on the accuracy of the classifier. This research focuses on SEMG feature extraction, which is applicable to hand movement recognition.

Many studies have attempted to extract a feature vector from SEMG signals using time or frequency analysis methods. For example, the EMG amplitude, root mean square, zero-crossing, autoregressive coefficients, Fourier transform coefficients and cepstrum coefficients have been used as components of the feature vector [8,9,10,11,12,13]. Angkoon et al. introduced thirty-seven state-of-the-art EMG feature extraction methods for the time or frequency domain and compared their classification accuracies [14]. These introduced methods, based on time or frequency analysis, have been widely used for prosthetics, but they cannot combine the time and frequency domains; thus, the extracted features contain limited information. Recent studies have shown that time-frequency analysis methods can extract more information about SEMG signals, such as short-time Fourier transform [15], wavelet transform [16], and wavelet packet transform [17], but they yield a high-dimensional feature vector, which increases the learning parameters of a classifier. Feature reduction can increase the calculation speed of the classifier. Several feature reduction methods, such as principal component analysis (PCA) [18], simple Fisher linear discriminant analysis, and fuzzy logic, have been applied to reduce the number of features. Englehart et al. extracted a time-frequency feature vector through a wavelet packet and used PCA to reduce the number of features [19]. However, the short-time Fourier transform has a fixed window width when the window function is selected; thus, it only has a single time-frequency resolution. The wavelet transform can adaptively change the window function, but the signal analysis ability is seriously affected by noise. The Stockwell transform (S-transform) is a time-frequency analysis method, and it is developed from the short-time Fourier transform and wavelet transform. This method can adaptively adjust the window function and avoid the defect of a fixed window width. Furthermore, the S-transform has characteristics that are insensitive to noise. The S-transform has been used in the analysis of EEG and exhibits excellent performance in removing ECG artifacts from EEG data [20,21,22]. To the authors’ knowledge, this technique has not been applied in EMG classification. In this research, the S-transform is used to create a feature vector, aiming to improve the recognition accuracy of the classifier.

The classification maps the feature vectors from extracted features into specific classes of motion. With the development of prosthetic hands with multiple degrees of freedom, it has become more effective to apply pattern-recognition-based control, which consists of feature extraction and classification during signal processing [23]. Support vector machines [24,25], Bayesian classifiers [26], evidence accumulation [27], fuzzy logic [28], Gaussian mixture model classifiers [29], and k-nearest neighbor classifiers [30] are the most common classification techniques. Recently, many studies have shown the success of neural networks and their ability to learn the distinction between different conditions in pattern recognition. In this research, an artificial neural network (ANN)-based multilayer perceptron (MLP) is used as a classifier to recognize hand movements. To evaluate the performance of the S-transform, three different methods are used to extract features, and these methods used the same classifier.

In this research, a novel feature extraction method is proposed to recognize five hand motions plus a rest state. This method is compared with state-of-the-art feature extraction methods, including wavelet transform and power spectral density. An evaluation using statistical criteria and measurement of classification accuracy are proposed to evaluate these three feature extraction methods. This paper proposes using the S-transform method to extract a feature vector and uses the PCA method to reduce the dimension of the feature. An ANN-based MLP is used as a classifier to recognize the hand motions (see Figure 1). This paper is organized as follows. Experiments and data acquisition are presented in Section 2. Section 3 presents a novel feature extraction method and the design of a classifier. The results and discussion are shown in Section 4 and Section 5, and finally, the conclusion is presented in Section 6.

## 2. SEMG Data Acquisition

In this research, forearm SEMG signals were collected with a BioRadio (wireless physiological collector from Great Lakes NeuroTechnologies company). Eight subjects, consisting of three amputees and five healthy subjects, participated in this experiment (see Figure 2). The basic information of the subjects is shown in Table 1. Two EMG acquisition channels (CH1 and CH2) were used to collect the SEMG signals. The two channels of surface electrodes were placed on the flexor carpi radialis muscle (CH1) and extensor carpi radialis longus muscle (CH2). For the amputees, the two electrode placements are the remnants of the two muscles. Each channel has two pairs of surface electrodes, and each electrode was separated from the other by 2 cm.

Before the data acquisition began, each subject was given a thorough written and oral explanation of the experiment itself, including the associated risks, the subjects would then sign an informed consent form. The experiment was approved by the Local Ethics Committee of Peking University.

Generally, the dominant energy of SEMG is concentrated in the range of 20–500 Hz, and its amplitude is limited to 0–10 mV. Therefore, the sampling frequency was set at 1000 Hz, and a bandpass filter of 20–500 Hz bandwidth was used. Each subject performed six hand motions: wrist flexion, wrist extension, hand open, hand close, key grip and rest state. The amputees imagined performing the same motions with the amputation side as with the sound side. The tests were repeated six times for each subject, resulting in 10 s of EMG signals per subject for each motion. The first three times were used for the learning procedures, and the remaining three times were used for the evaluation of recognition performance. Fatigue of the subjects was avoided with a 2-minute resting period after each exercise. In Figure 3, the raw surface EMG data of six motions in a 256-point window from subject 1 collected by CH1 are shown. The window size of the EMG sample was set to 250 ms (256 samples) for real-time control to guarantee a response time of less than 300 ms, which allows the users to operate the prosthetic hands without perceiving a time delay.

## 3. Algorithm Description

The classification of hand motions through the SEMG signal includes three main steps: preprocessing, feature extraction and pattern recognition. The preprocessing of SEMG signals was performed using hardware. The novel feature extraction with the S-transform was proposed, and four selected features were proposed for one channel after the S-transform. To speed up the computation, the PCA method was adopted to reduce the dimensions. The details of the design process of the ANN-based MLP were introduced, which include the structure of the ANN model and loss function calculation method.

### 3.1. Feature Extraction and Reduction

The S-transform is developed from the short-time Fourier transform and wavelet transform, and it was proposed by Stockwell in 1996 [21]. The S-transform adopts a Gaussian window function, and the window width is proportional to the inverse of the frequency, which avoids the defect of a fixed window width. In addition, the extracted feature used in the S-transform method is insensitive to noise. The S-transform of signal h(t) is
(1)$$S\left(\tau ,f\right)=\int_{-\infty }^{\infty }h\left(t\right)\frac{\left|f\right|}{\sqrt{2\pi }}e^{-\frac{{\left(\tau -t\right)}^{2}f^{2}}{2}}e^{-i2\pi ft}dt\;$$
where *t* and *f* represent the time and the frequency, respectively. *i* is the imaginary unit, and τ is the center of the Gaussian window function. Equation (Equation 1) shows that the width and height of the Gaussian window vary with frequency. Let
(2)$$a\left(t,f\right)=h\left(\tau \right)e^{-i2\pi f\tau },b\left(\tau -t,f\right)=\frac{\left|f\right|}{\sqrt{2\pi }}e^{-\frac{{\left(\tau -t\right)}^{2}f^{2}}{2}}$$

From Equation (Equation 2), the S-transform of signal h(t) can also be shown as the following equation:(3)$$S\left(\tau ,f\right)=\int_{-\infty }^{\infty }a(t,f)b(\tau -t,f)dt$$
C(α,f) is a Fourier transform of S(τ,f) (from τ to α), and A(α,f) and B(α,f) are the Fourier transforms of a(t,f) and b(τ,f) (from τ to α ), according to the convolution theorem:(4)C(α,f)=A(α,f)B(α,f)
(5)$$C(\alpha ,f)=H(\alpha +f)e^{-\frac{2\pi \alpha^{2}}{f^{2}}}$$
H(α+f) is the Fourier transform of a(t,f), the exponential term is the Fourier transform of b(τ,f), and the S-transform can be calculated via inverse Fourier transform by Equation (Equation 5) (from α to τ):(6)$$S\left(\tau ,f\right)=\int_{-\infty }^{\infty }H\left(\alpha +f\right)e^{-\frac{2\pi \alpha^{2}}{f^{2}}}e^{i2\pi \alpha \tau }d\alpha$$

Let *f*→n/NT and τ→jT; the discrete S-transform can be shown as the following equations:(7)$$\left\{\begin{array}{c} S\left[jT,\frac{n}{NT}\right]=\sum_{m=0}^{N-1}H\left[\frac{m+n}{NT}\right]e^{-\frac{2\pi^{2}m^{2}}{n^{2}}}e^{\frac{i2\pi mj}{N}},n\neq 0\\ S\left[jT,0\right]=\frac{1}{N}\sum_{m=0}^{N-1}h(\frac{m}{NT}),n=0 \end{array}\right.$$
(8)$$H\left[k\right]=\frac{1}{N}\sum_{m=1}^{N-1}h\left(m\right)e^{-\frac{i2\pi km}{N}}$$
where *j*, *m*, and n=0,1,…,N−1. As shown in Equation (Equation 8), the H[k] is discrete transform of time series h(m). From Equation (Equation 7), the results of the discrete S-transform is a two-dimensional matrix, which is named the S-matrix. The raw elements of the S-matrix represent the frequency, and the column elements represent the sampling time. The S-matrix can be calculated according to Equation (Equation 7) using the fast Fourier transform and the inverse Fourier transform.

The raw signals were segmented into a 256-point window, and a signal with 256 points was defined as a segmented signal. Therefore, a matrix of segmented signals (2 × 256 matrix) can be one input for the control system of the prosthetic hand. The raw EMG signal can be collected as a two-dimensional matrix of time amplitude, and the S-transform can transform it into a three-dimensional matrix of time-frequency-amplitude.

The calculations of four features extracted from the S-matrix are studied in this research: (1) Maximum frequency in one segment signal (256 points) F1; (2) The amplitude of the maximum frequency in one segment signal (256 points) F2; (3) The median frequency in one segment signal F3; (4) The energy of one segment signal F4. The other channel of the feature vector is defined as F(F5,F6,F7,F8). Due to the two channels of EMG signals, the feature vector becomes eight dimensional (see Figure 4).

Next, the PCA method is used for feature reduction, which can improve the speed of the classifier. As shown in Figure 4, PCA preforms dimensionality reduction for two channel features, and after reduction by PCA, the number of features is reduced from eight to three. PCA has the advantage of automatically ranking the importance of features in the projection space. Then, the three features were used as an input vector for pattern recognition.

### 3.2. Classification

Finally, an ANN is used as the classifier. The ANN has one hidden layer with 10 neurons (see Figure 5). The network structure was determined by trial and error. The forward-propagating order is: linear function, ReLU function, linear function and softmax function. This process can be represented by the following equations:(9)$$z^{\left[1](i\right)}=W^{\left[1\right]}x^{\left(i\right)}+b^{\left[1](i\right)}\;$$
(10)$$a^{\left[1](i\right)}=relu\left(z^{\left[1](i\right)}\right)\;$$
(11)$$z^{\left[2](i\right)}=W^{\left[2\right]}a^{\left[1](i\right)}+b^{\left[2](i\right)}\;$$
(12)$$y_{a}=\sigma \left(z^{\left[2](i\right)}\right)=\frac{e^{Z^{\left[2](i\right)}}}{\sum_{K=1}^{K}e^{Z_{k}}}\;$$
where $x^{\left(i\right)}$ is the input vector, $W^{\left(1\right)}$ and $W^{\left(2\right)}$ are weight parameters, and $b^{\left[1](i\right)}$ and $b^{\left[2][i\right]}$ are bias parameters. σ represents the softmax regression function, and *k* represents the labels. i=0,1,…,10,k=0,1,…,6. $y_{a}$ represents the probability of the *k*th motion. The cost function *J* can be computed as follows:(13)$$J=\sum -{\hat{y}}_{b}Iny_{b}$$
where $\hat{y}{}_{b}$ is the truth value of the *i*th category and $y_{b}$ is the prediction results of the output. The advanced optimization method is used for the backpropagation of the neural network, and our final target is minimizing the cost function *L*. The PCA-reduced feature vector is used as the input vector, and after training the neural network, the optimized parameter can be determined. Each label maps one hand motion, and the labels turn to one-hot encoding for the data processing (see Figure 5). This classifier uses the adaptive moment estimation (Adam) optimization method, which is an adaptive learning rate method. This method can be described by the following equations for l=1,…,L:(14)$$\left\{\begin{array}{c} v_{dW^{\left[l\right]}}=\beta_{1}v_{dW^{\left[l\right]}}+\left(1-\beta_{1}\right)\frac{\partial J}{\partial W^{\left[l\right]}}\\ v_{dW^{\left[l\right]}}^{corrected}=\frac{v_{dW^{\left[l\right]}}}{1-{\left(\beta_{1}\right)}^{t}}\\ s_{dW^{\left[l\right]}}=\beta_{2}s_{dW^{\left[l\right]}}+\left(1-\beta_{2}\right){\left(\frac{\partial J}{\partial W^{\left[l\right]}}\right)}^{2}\\ s_{dW^{\left[l\right]}}^{corrected}=\frac{s_{dW^{\left[l\right]}}}{1-{\left(\beta_{1}\right)}^{t}}\\ W^{\left[l\right]}=W^{\left[l\right]}-\alpha \frac{v_{dW^{\left[l\right]}}^{corrected}}{\sqrt{s_{dW^{\left[l\right]}}^{corrected}}+\varepsilon } \end{array}\right.$$

It calculates an exponentially weighted average of past gradients and stores it in variables *v* and $v^{corrected}$. Variables *s* and $s^{corrected}$ were used to store the average weights of the squares of past gradients. Here,

t counts the number of steps taken by the Adam optimizerL is the number of layers$\beta_{1}$ and $\beta_{2}$ are hyperparameters that control the two exponentially weighted averages; generally, $\beta_{1}$ = 0.9, and $\beta_{2}$ = 0.999α is the learning rateε is a very small number to avoid dividing by zero; generally, ε = 10E−8

The ANN model parameters were determined by the training data when the cost function *J* reached the minimum value, and the accuracy of the classifier in the test set was obtained, which can evaluate the performance of feature extraction methods.

## 4. Results

This research compared three feature extraction methods: power spectral density, wavelet packet transform and the proposed S-transform. Power spectral density and wavelet packet transform are popular advanced methods in EMG feature extraction. From the statistical criterion method and classifier measurement, the best feature selection method can be obtained.

Power spectral density is a frequency-domain analysis method. The mean frequency, median frequency and peak frequency are selected as features via the power spectral density. Each feature was calculated by 256 points. The mean frequency $F_{a}$, median frequency $F_{b}$, and peak frequency $F_{c}$ are shown as the following equations:(15)$$F_{a}=\sum_{j=1}^{M}f_{j}P_{j}/\sum_{j=1}^{M}P_{j}$$
(16)$$\sum_{j=1}^{F_{b}}P_{j}=\sum_{j=F_{b}}^{M}P_{j}=\frac{1}{2}\sum_{j=1}^{M}P_{j}$$
(17)$$F_{c}=\max\left(P_{j}\right),j=1,\ldots ,M\;$$
where $f_{j}$ is the frequency of the spectrum at frequency bin *j*, $P_{j}$ is the EMG power spectrum at frequency bin *j*, and *M* is the length of the frequency bin. After the PCA method, the six features from the two channels are reduced to three. The PCA-reduced features are used as an input vector for the ANN-based MLP.

The wavelet packet transform is a time-frequency analysis method. Englehart. K. used the symmlet mother wavelet of order five as the wavelet packet basis function, and the results showed that it has a good effect on the EMG signal process [19]. Three levels of wavelet decomposition and symmlet mother wavelet of order five were adopted in this research. The original signal is *S*, and a decomposition of *S* into binary-tree-structured subspaces using the wavelet packet transform is shown in Figure 6. In this study, the maximum wavelet packet coefficient was chosen as the EMG feature. This was explained by the following equation:(18)$$F=\{\max\left|S_{i}\right|,i=1,\ldots ,2^{level}\}$$
where $S_{i}$ are wavelet coefficients after the decomposition, level = 3, and *F* represents the feature vector. Finally, the features were reduced by the PCA method in the same manner as the S-transform.

### 4.1. Evaluation of the S-transform Method

To evaluate the classification performance, the scatter plots of EMG features using the S-transform method are shown in Figure 7a, which are compared with the scatter plots of EMG features using the wavelet transform (see Figure 7b) and power spectral density (see Figure 7c). There are three features after feature reduction. The scatter plots of each of the two features are shown from six movements in three methods. Each movement uses a specific symbol and color with 100 sample points. Class separability can evaluate the performance of EMG features. The good quality in class separability indicates that the highest separation between classes is obtained. Generally, the statistical criterion method was used to evaluate the distance between two scatter groups and directly address the variation of features in the same group.

The RES index [31] uses the Euclidean distance as a distance function and the standard deviation as a dispersion measure. Euclidean distance is calculated as the root of square differences between coordinates of a pair of objects, and standard distance is the most robust and widely used measure of the variability. The best extraction performance is obtained when the Euclidean distance value is high and the standard deviation value is low.

The RES index uses the Euclidean distance as the numerator and the standard deviation as the denominator. Thus, a larger RES index signifies a higher class separation of each class. The RES(p,q) index of two motions (*p* and *q*) can be expressed as
(19)$$ED\left(p,q\right)=\sqrt{{\left(p_{1}-q_{1}\right)}^{2}+{\left(p_{2}-q_{2}\right)}^{2}}$$
(20)$$\sigma =\sqrt{\frac{\sum_{w=1}^{N_{w}}{\left(r_{w}-\mu \right)}^{2}}{N_{W}}}$$
(21)$$RES\left(p,q\right)=\frac{ED(p,q)}{\bar{\sigma }}\;$$
(22)$$F_{norm}=\frac{r+\min(r)}{\max(r+\min(r)}s$$
where *p* and *q* are the feature means of two motions from six hand motions with two channels. *r* is the feature of the $w^{th}$ (*w* = 100) window of $N_{w}$, and σ is the feature mean of all windows. $\bar{\sigma }$ is the average between the standard deviation of two motions (p and q) with two channels. In addition, the EMG features are normalized before calculating the RES index. The normalization of features $F_{norm}$ is performed, which can be expressed as Equation (Equation 22).

The average of RES(p,q) from fifteen possible combinations of six hand movements was defined as the RES index. From the results, the RES indices of features using the proposed S-transform method are 9.14, 6.98, and 7.16, which are higher than the RES indices using wavelet transform and power spectral density, which means that the S-transform has the best quality of class separability. The results of the evaluation indicate that the S-transform is an effective feature extraction method.

### 4.2. Results of the Classifier

With the training data, the parameters of the ANN-based MLP were determined in the learning procedure. The recognition accuracies of the classifier using different feature extraction methods with test data are shown in Figure 8. The S-transform features provide 97.46%, 97.63%, 97.98%, 98.83%, 98.65%, 98.21%, 98.3%, and 97.91% mean accuracy in each subject. The wavelet packet transform provides 96.82%, 97.32%, 97.45%, 98.21%, 97.96%, 97.88%, 97.76%, and 97.44% mean accuracy, and the power spectral density provides 93.62%, 94.23%, 94.86%, 96.71%, 96.9%, 96.88%, 97.11%, and 97.21% mean accuracy in each subject. The results that used the proposed feature extraction method with four subjects showed higher accuracy than the wavelet transform method and power spectral density method.

In Figure 9, the red line represents the S-transform, the blue line represents the wavelet packet transform, and the green line represents the power spectral density. The average accuracy was improved by increasing the number of features with the three methods. When the number of features is equal to three, the speed of the increase becomes very slow. Comprehensively considering the performance and application, the three features will be more effective for the EMG classification method. With different numbers of features, the S-transform is superior to the wavelet transform and power spectral density. Thus, the S-transform can improve the class separability and classification accuracy.

## 5. Discussion

In this paper, the authors have proposed the S-transform as an efficient method for SEMG feature extraction. To the authors’ knowledge, this is the first study in which this algorithm has applied for hand movement recognition. Feature extraction is an important step in myoelectric pattern recognition, and the proposed method shows excellent performance in extracting features and classifying the SEMG data for different hand movements compared with other methods.

Previous studies have shown method based on the wavelet transform and the power spectral density have achieved competitive accuracy for classifying hand gestures compared with other existing methods [3,14]. Although its performance appeared to be increased by only a few percent (see Figure 8), the S-transform method was superior to the wavelet transform and power spectral density methods under the controlled conditions of this study, as proven using statistical methods. In many studies, the number of electrodes is increased to improve the classification accuracy [32,33,34]. However, the usage of multiple EMG sensors on a prosthetic hand is not practical. First, the weight of the socket will increase, causing an amputee to feel discomfort when wearing the SEMG sensor array. Second, as the number of electrodes increases, more dimensions of data must be processed, and the amount of computation will also increase. Thus, commercial advanced prosthetics typically have only one or two electrodes [35]. This paper therefore considers two channels, consistent with typical prosthetic hand applications.

Many studies have identified more than 10 distinct hand movements and have achieved up to 80–90% accuracy. In this paper, the authors use 2 electrodes to distinguish 6 hand movements with up to 95–98% under controlled conditions. The ability to recognize 6 hand movements for a prosthetic hand is superior to the capabilities of current commercial intelligent prostheses. Recognizing a larger number of hand movements would hinder practical application due to issues related to computational complexity and practicality.

In this paper, a three-layer ANN algorithm has been adopted as the classifier to evaluate the quality achieved with different numbers of features. In designing the classifier, a linear softmax model is used here to learn a good distribution of factorized features. Although a convolutional neural network (CNN) could offer improved accuracy for hand movement classification [36], a large amount of data would be required for pretraining. Because a CNN has many layers, its calculation time is long. Therefore, an ANN was instead adopted in this paper to evaluate the proposed feature extraction method. The proposed time-frequency method requires feature reduction due to the high dimensionality of the data; therefore, choosing an optimal number of features is very important. As seen from Figure 8, there are considerable changes after dimensionality reduction to between 1 dimension and three dimensions, whereas the change in accuracy with more than three dimensions is stable. Therefore, 3-dimensional features are chosen in this paper.

After more than 70 years of development of EMG signal control methods, the main approaches are still dominated by threshold control. The main reason is that EMG signals are nonlinear and random, making it very difficult to extract characteristic quantities. The superiority of the S-transform has been verified offline in this paper. In the future, this method could be used to control various movements of prosthetic limbs in an online manner. In practical applications, the characteristics of different movements may change with muscle fatigue; in such a case, the parameters of the neural network will need to be relearned to adjust to these changes. If the classifier can learn autonomously, then the algorithm proposed in this paper, combined with an artificial intelligence algorithm, will be very effective in clinical application.

## 6. Conclusions

A method was proposed to extract forearm EMG signal features in the time-frequency domain using the S-transform. The dimension of the S-transform features was then reduced by PCA. Subsequently, an ANN-based MLP was proposed to classify the six hand motions. According to the results of the RES index, the proposed method has the best overall performance compared with current popular feature extraction methods. The experimental results show that the S-transform can offer more information about the EMG signal, which can improve the class separability and classification accuracy. In future work, this method could be used for prosthetic hand control, which could improve the accuracy of hand motion classification.

## Figures and Tables

**Figure 1 sensors-19-04457-f001:**
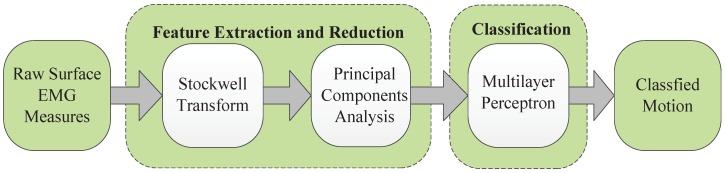
Block diagram for EMG pattern recognition.

**Figure 2 sensors-19-04457-f002:**
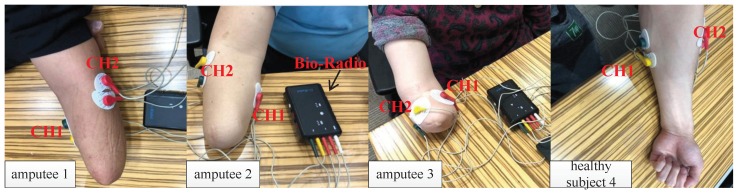
EMG data acquisition from some of the subjects.

**Figure 3 sensors-19-04457-f003:**
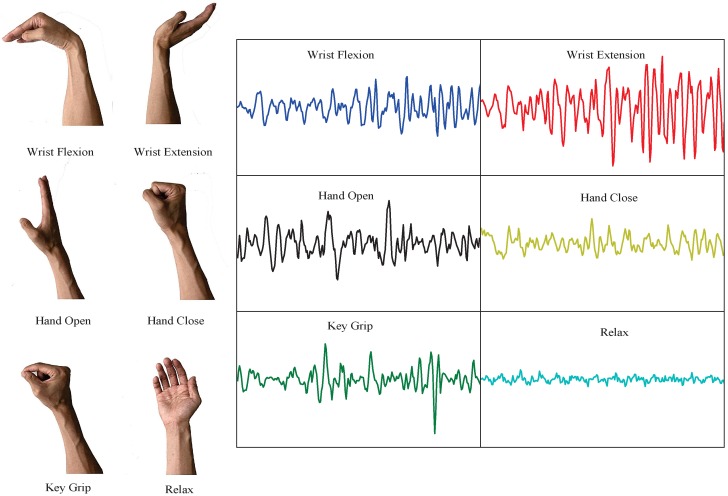
Raw EMG in 256-point window of one subject performing six motions by CH1.

**Figure 4 sensors-19-04457-f004:**
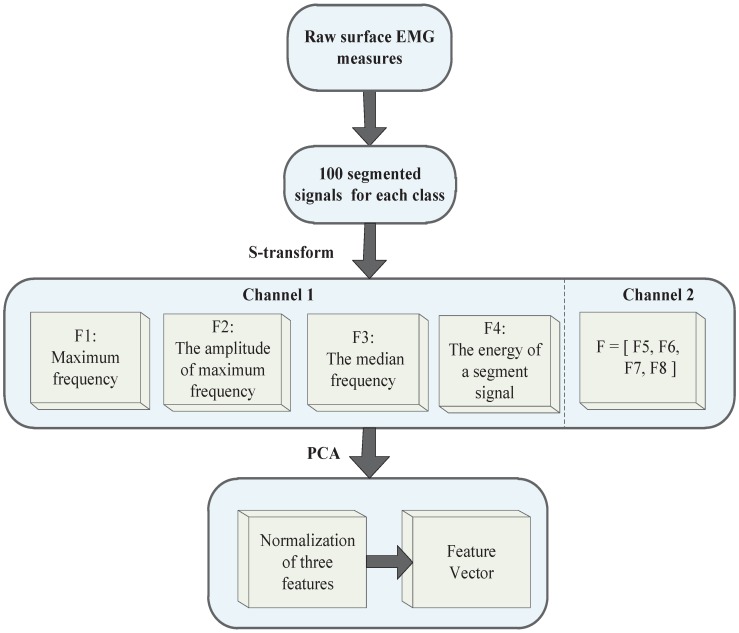
Feature extraction algorithm.

**Figure 5 sensors-19-04457-f005:**
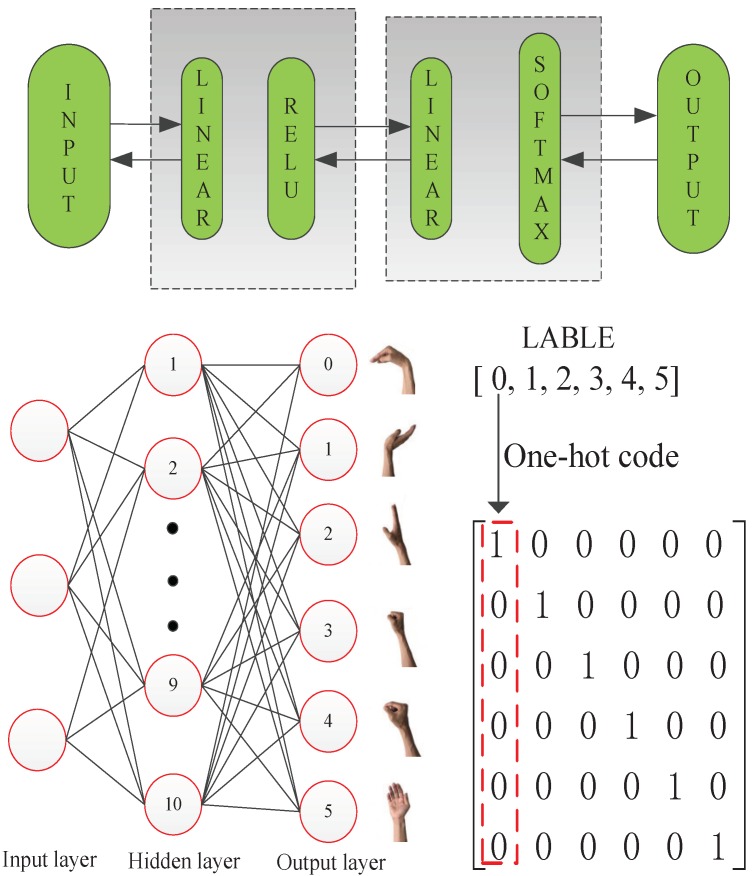
ANN-based multi-layer perceptron.

**Figure 6 sensors-19-04457-f006:**
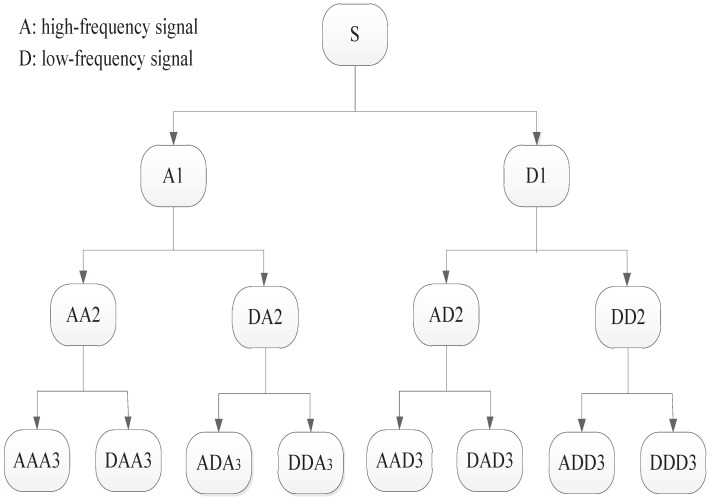
Wavelet transform decomposition tree from decomposition level 3.

**Figure 7 sensors-19-04457-f007:**
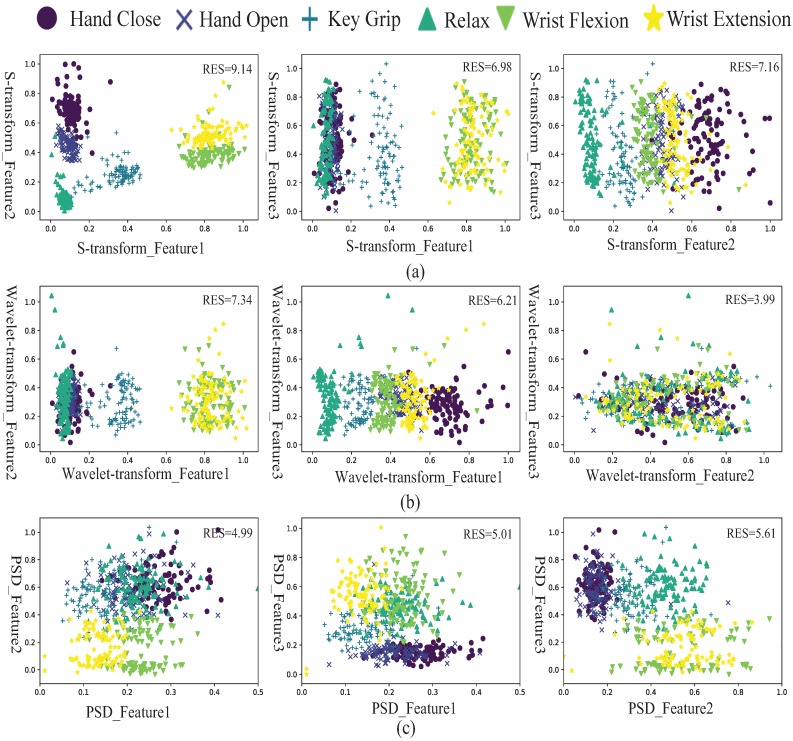
(**a**) Scatter plot of six different movement features extracted using the S-transform. (**b**) Scatter plot of six different movement features extracted using the wavelet transform. (**c**) Scatter plot of six different movement features extracted using the power spectral density.

**Figure 8 sensors-19-04457-f008:**
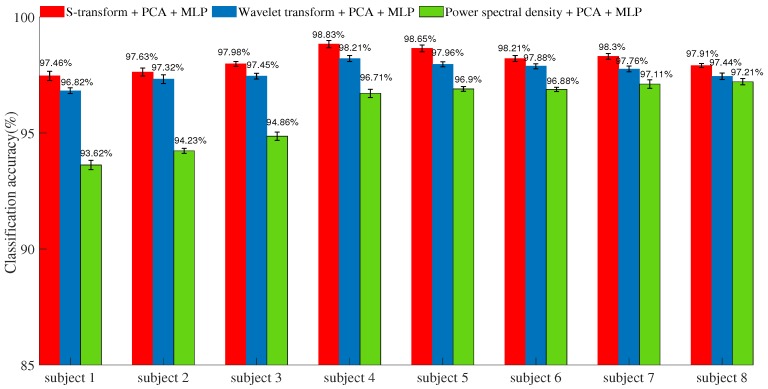
Classification accuracies for different feature methods.

**Figure 9 sensors-19-04457-f009:**
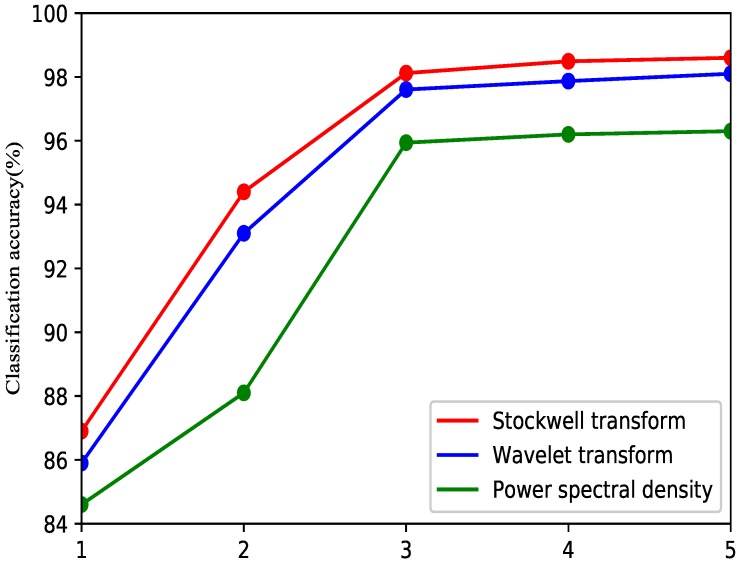
Average classification accuracies for Stockwell transform, wavelet transform features and power spectral density.

**Table 1 sensors-19-04457-t001:** The basic information of the subjects.

Subject	Gender	State	Age (Years)	The Time of Amputation (Years)
1	male	amputee	58	2
2	male	amputee	56	30
3	female	amputee	55	35
4	male	healthy	29	/
5	male	healthy	32	/
6	female	healthy	27	/
7	male	healthy	33	/
8	male	healthy	35	/

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
