# Peer review of "SEMG Feature Extraction Based on Stockwell Transform Improves Hand Movement Recognition Accuracy"

_sensors, 2019, doi:10.3390/s19204457_

Round 1
Reviewer 1 Report
The title of this article is “SEMG feature extraction...”, so there are comparisons between S-tranform, wavelet transform and power spectrum based features, and after ANN, to distinguish the resolution of the six hand gestures.
The topic is very interesting and the algorithm is also very interesting. Only the object being tested is Amputee. This is another topic. It may be named as” Amputee imagined EMG signal analysis based on S-tranform”. This article deals with two issues. It’ s confusing.
If the focus is on the first one, author can use the data of healthy people, and even used open database of hand gesture to test S-transform; if the focus is on the second one, the number of experiments should be increase, and most important of all, the experiment should be taken with certified IRB.
Author Response
The comments are all valuable and have been very helpful in revising and improving our paper as well as in providing important guiding significance to our research.
Point 1 : Only the object being tested is Amputee. This is another topic. It may be named as” Amputee imagined EMG signal analysis based on S-transform”. This article deals with two issues. It’ s confusing.
If the focus is on the first one, author can use the data of healthy people, and even used open database of hand gesture to test S-transform; if the focus is on the second one, the number of experiments should be increase, and most important of all, the experiment should be taken with certifie IRB.
Respond 1:First, our experiment was approved by the Local Ethics Committee of Peking University; the IRB number is 2018-06-02, and we have sent a copy of the ethical approval file to the journal mailbox. We have also added a corresponding explanatory statement in the paper.
Although the amputees imagined performing the motions, they generated real EMG signals. Because of the subject dependency of EMG signals, data from healthy people are valuable as a reference for verifying the robustness of the algorithm. However, the final goal of the paper is to apply the S-transform algorithm to prosthetic hands; therefore, amputees also participated in this experiment.
For example, [1] proposed the use of EMG data for noninvasive naturally controlled robotic hand prostheses. Both healthy subjects and amputees participated in the experiment.
Finally, the authors have added five healthy subjects to increase the number of experiments. The new results are shown in the paper.
[1] Atzori, M., Gijsberts, A., Castellini, C., Caputo, B., Hager, A. G. M., Elsig, S., ... & Müller, H. (2014). Electromyography data for non-invasive naturally-controlled robotic hand prostheses. Scientific data, 1, 140053

Reviewer 2 Report
This paper illustrates a novel method for classifying hand grasp related EMG patterns, and compares the performance of the classifier with other methods available in the literature.
While the potential of the novel algorithm for pattern recognition may be interesting for the literature, as it shows very good performances, I suggest that this work is published after careful revisions related to its presentation, framing within the state of the art, and description of the results.
First, my opinion is that, in general, the paper is lacking some references to the existing literature, both in the state-of-the-art and discussion sessions. This is also reflected in the very poor discussion section, that is merged with results (this should be modified), rather than opening a broader view to existing works. For example, the Ninapro database (1) uses 8 to 14 EMG electrodes for similar applications, analyzing forearm patterns at several levels: (algorithms for pattern identification (2 and many others) or with other EMG feature extracting methods such as (3). Is it adequate to use less electrodes? This should be commented. Check that you have included all the literature in the topic, such as: 4,5,6,7, that are milestones you should not miss that help to comment results and better explain the rationale of your work. Please complete the screening and check if I missed relevant references.
All your results should be commented in the light of what previous authors have found in the topic, justifying several aspects of your study design: reduced number of EMG channels (if this is an advantage in some way, please discuss it), low number of subjects (especially healthy controls), adoption of a novel method. Especially how these results could be translated to a higher number of subjects, and on which applications, describing the impact. All this should be commented in much more detail.
There are several study designs that use different approaches for classification, such as employing more EMG channels. You should discuss in more detail if your approach has comparable accuracies.
I expect that the method session reports with more clarity that this paper is comparative in respect to state-of-the-art algorithms, and also the outcome measures should be clarified better.
Are there devices or applications that may benefit from your work? Why?
Figures: it is my opinion that the figures are vertically stretched and may look better if they were not. So, I strongly suggest to improve them, especially in Figure 7 where some captions are hardly readable.
1 Atzori, M., Gijsberts, A., Castellini, C., Caputo, B., Hager, A. G. M., Elsig, S., ... & Müller, H. (2014). Electromyography data for non-invasive naturally-controlled robotic hand prostheses. Scientific data, 1, 140053
2 Atzori, M., Cognolato, M., & Müller, H. (2016). Deep learning with convolutional neural networks applied to electromyography data: A resource for the classification of movements for prosthetic hands. Frontiers in neurorobotics, 10, 9
3 Scano, A., Chiavenna, A., Molinari Tosatti, L., Müller, H., & Atzori, M. (2018). Muscle synergy analysis of a hand-grasp dataset: a limited subset of motor modules may underlie a large variety of grasps. Frontiers in neurorobotics, 12, 57.
4 Li, G., & Kuiken, T. A. (2009, September). EMG pattern recognition control of multifunctional prostheses by transradial amputees. In 2009 Annual International Conference of the IEEE Engineering in Medicine and Biology Society (pp. 6914-6917). IEEE.;
5 Jiang, N., Vest-Nielsen, J. L., Muceli, S., & Farina, D. (2012). EMG-based simultaneous and proportional estimation of wrist/hand kinematics in uni-lateral trans-radial amputees. Journal of neuroengineering and rehabilitation, 9(1), 42.
6 Reilly, K. T., Mercier, C., Schieber, M. H., & Sirigu, A. (2006). Persistent hand motor commands in the amputees' brain. Brain, 129(8), 2211-2223.
7 Naik, G. R., Al-Timemy, A. H., & Nguyen, H. T. (2015). Transradial amputee gesture classification using an optimal number of sEMG sensors: an approach using ICA clustering. IEEE Transactions on Neural Systems and Rehabilitation Engineering, 24(8), 837-846.
Author Response
Thank you for your advice. The comments are all valuable and have been very helpful in revising and improving our paper as well as in providing important guiding significance to our research.
Point.1:
In general, the paper is lacking some references to the existing literature, both in the state-of-the-art and discussion sessions. This is also reflected in the very poor discussion section, that is merged with results (this should be modified), rather than opening a broader view to existing works. For example, the Ninapro database (1) uses 8 to 14 EMG electrodes for similar applications, analyzing forearm patterns at several levels: (algorithms for pattern identification (2 and many others) or with other EMG feature extracting methods such as (3). Is it adequate to use less electrodes? This should be commented. Check that you have included all the literature in the topic, such as: 4,5,6,7, that are milestones you should not miss that help to comment results and better explain the rationale of your work. Please complete the screening and check if I missed relevant references.
All your results should be commented in the light of what previous authors have found in the topic, justifying several aspects of your study design: reduced number of EMG channels (if this is an advantage in some way, please discuss it), low number of subjects (especially healthy controls), adoption of a novel method. Especially how these results could be translated to a higher number of subjects, and on which applications, describing the impact. All this should be commented in much more detail.
I expect that the method session reports with more clarity that this paper is comparative in respect to state-of-the-art algorithms, and also the outcome measures should be clarified better.
Response1:
First, the authors have modified the structure of the paper by adding a discussion section, and some important citations have been added in both the state-of-the-art and discussion sections. As the expert noted, it is necessary to comment on whether it is adequate to use so few electrodes. Therefore, we have explained why we consider only two channels in the discussion section.
Second, the authors have added five healthy subjects to increase the number of experiments. The final results have been revised accordingly, and some figures and tables have been changed. The authors have read the listed references; in particular, [5] reports the control of two or three degrees of freedom simultaneously. This is another topic of interest for EMG control. The authors have added the relevant reference to support our comments on the results and better explain the rationale of our work.
Point.2: Are there devices or applications that may benefit from your work? Why?
Respond2: Our team focuses on prosthetic hand application. We have developed three types of bionic hands: Types A, B, and C. Type A has one degree of freedom and is controlled by the EMG amplitude threshold. Type B has two degrees of freedom, and FFT and ANN algorithms are used to control it. Type C is a multi-degree-of-freedom hand, and we wish to use the proposed S-transform and ANN algorithms to control various movements of the prosthesis based on SEMG data. In this paper, we have completed the necessary offline processing. In our future work, we will attempt to apply the algorithm for online control.
Point.3:Figures: it is my opinion that the figures are vertically stretched and may look better if they were not. So, I strongly suggest to improve them, especially in Figure 7 where some captions are hardly readable.
The authors have adjusted the heights of the figures and believe that the layout of the paper looks better now. In Figure 7, the captions have been changed for better ease of reading.

Round 2
Reviewer 1 Report
The author had responded my previous comments.
Reviewer 2 Report
The authors have improved the quality of the paper according to the suggestions.